# Exploring attitudes to decolonising the science curriculum—A UK Higher Education case study

Lena Grinsted[1]*, Catherine Murgatroyd[2], Jodi Burkett[3]

1 School of the Environment and Life Sciences, University of Portsmouth, Portsmouth, United Kingdom, 2 Department of Curriculum and Quality Enhancement, Academic Development, University of Portsmouth, Portsmouth, United Kingdom, 3 School of Area Studies, Sociology, History, Politics and Literature, Portsmouth, United Kingdom

* lena.grinsted@port.ac.uk

**Data Availability Statement:** All relevant data are within the article and its Supporting information files.

**Funding:** The author(s) received no specific funding for this work.

## Abstract

Scientific advances are historically linked to colonial actions of past empires resulting in knowledge production biased towards the West with minimal representation of scholars of other ethnicities than White in science curricula in Higher Education (HE). Calls to decolonise science curricula seek to diversify content by acknowledging the role of racism and privilege in the history of science, aiming at creating a HE that is less isolating for minoritised ethnicities and feels welcoming to students of all identities. This case study explored science teaching staff's familiarity with and misconceptions of decolonisation at a UK HE institution using an online questionnaire. We further assessed participants' perceptions of barriers, benefits and risks, training needs, and preparedness to take actions in their teaching. We found that a majority of participants had a positive disposition towards decolonising their teaching, but that critical misconceptions, e.g. linking decolonisation to 'cancel culture' and 'colour-blind' behaviour were common, while important barriers, e.g. a lack of training and constraints on time, halt progress. We provide specific recommendations for staff training and a brief historical background relevant to life sciences. By supporting teachers, that train future generations of scientists, to decolonise the curriculum we can improve equity in HE, academia, and society.

## Introduction

Recognised scientific practises are shaped by ever-changing paradigms that are intimately linked with constantly changing socio- and geo-political landscapes [1–3]. Major scientific developments have, through history, often been motivated by political, religious, or moral convictions of the time, and in particular the colonial expansions of British and other European empires have had substantial impacts on scientific knowledge production, dissemination, and access to that knowledge [2, 4, 5]. In this way, colonial legacies contribute to current global inequalities on multiple axes: strong biases still exist based on heritage, citizenship, gender, sexual orientation, (dis)ability—and more—in access to, and progression within, higher

**Competing interests:** The authors have declared that no competing interests exist.

education and academia, and in representation in scientific publications and science curricula [1–3, 6]. Decolonisation of scientific practises, as one strand of global Decolonisation, aims to break down existing structures that perpetuate inequalities and privilege, to ensure that all demographics are included and represented, that historical legacies are acknowledged, and that science is accessible to all [2–5, 7, 8]. However, to successfully decolonise scientific practises in the longer term, the science curriculum used to educate and train future generations of scientists must also undergo decolonisation. This is a major undertaking that requires science teaching staff at Higher Education (HE) institutions to be familiar with the past and current intimate links between inequity and science, while also feeling prepared and comfortable taking steps towards decolonising their teaching and learning content and delivery.

## Aims and scope

In this study we aimed to explore the disposition of HE teaching staff towards decolonising the science curriculum using a medium-large sized university in England as a case study. Using an online questionnaire, we investigated staff members' i) familiarity with and misconceptions of decolonisation; ii) preparedness to take actions towards decolonising their teaching and content delivery; iii) perceptions of the benefits and risks associated with decolonisation; iv) perceived barriers preventing them from decolonising their teaching; and v) perception of which areas of training in relation to decolonisation might be beneficial for themselves and others. We further tested whether age, gender and scientific discipline could predict various aspects of staff members' disposition towards decolonisation. We use the results to propose specific recommendations for relevant staff training. While a case study does not allow us to universalise our findings, we argue that our recommendations are relevant for similar institutions in the UK and beyond who are at a relatively early stage in undertaking a critical review of their curriculum through a decolonial lens.

In the next section we describe and define decolonisation in relation to UK HE science curricula. After that, we briefly outline the historical background to the need for decolonising the sciences. We do this, firstly, to describe the social, educational and scientific paradigm underpinning the development and construction of UK universities and Western knowledge production: a paradigm in which human beings were seen as unequal and qualitatively different; where characteristics including biological sex, skin colour, cultural norms, physical and/or intellectual abilities determined individual status and 'worth' in society. Secondly, we do this to emphasise how recent this paradigm was the norm in Western society, accepted, and championed by scientists, educators and politicians, and how deeply embedded structural inequality is within the fabric of UK HE institutions.

## Defining decolonisation

Decolonising the curriculum is one strand of socially just education that aims to create an inclusive and enabling learning space and education for all students, improving student experience, wellbeing and feelings of belonging. A decolonised science curriculum frames scientific advances in their historical context, while acknowledging the people(s) who have not been given appropriate credit for their contributions. In the process, the racial hierarchies upon which science has been built and the sometimes racist or discriminatory motivations behind scientific research themes are exposed. Such a curriculum is meant to critically address White privilege, entitlement, inequity and racialised disadvantage [9–11]. Hence, decolonisation involves recognising that colonialism and racism have been involved in shaping modern history and pedagogy, affecting–and at times directing–knowledge production and scientific discoveries [4, 6, 10, 12–14]. It is important to stress that the aim of decolonising the science

curriculum is addition rather than subtraction: to discuss the biases and inequalities associated with science production and to add context and historical background to scientific advances. The tabloid media in the UK are known to portray decolonising efforts as attempts to censor out seminal scientific discoveries credited to scholars with racist views or motivations. However, this is a misconception. The decolonisation literature clearly calls for explicitly acknowledging traditional scholars' privilege, biases and motivations rather than removing their contributions from the curriculum [1, 2, 5, 9, 15].

Decolonisation is meant to benefit all students, not just those who are historically marginalised, preparing everyone for a modern, multi-cultural world, where privilege and entitlement is increasingly challenged. Such efforts aim to ensure that all students see themselves in their education, and feel some level of ownership over the knowledge they learn, while educating everyone–no matter their genetic or cultural heritage–to the best of their abilities [9, 10]. It further improves literacy and awareness of global inequalities in scientific knowledge production and equips future generations of scientists with tools to conduct scientific research that is more inclusive, equitable and just. Current curricula can be considered unrepresentative and privileged, because they selectively construct teaching that exclude certain marginalised narratives and perpetuate advantage and progression of particular groups. Such curricula can feel inaccessible to students who cannot identify with the narratives presented, leading to feelings of isolation and a lack of belonging for students of marginalised identities [6, 16, 17]. Anti-racist and decolonising activities introduced into teaching and learning spaces can be rewarding and empowering for both students and educators. However, in order for efforts to be impactful and increase students' feelings of belonging and ownership, teaching staff need to feel comfortable discussing emotive concepts like racism and privilege and be equipped with skills and tools to navigate such sensitive topics [18–21].

Decolonisation and inclusive, equitable teaching practises are in focus at an increasing number of UK universities, but the process is slow and efforts vary greatly in their scope and impact [14, 21, 22]. This is due to multiple factors which include confusion about what decolonisation actually means, which terminology is appropriate to use, which actions are necessary and impactful, and what the success criteria look like [9]. Perceived links between decolonisation work and activism, controversy and radicalism might also be a factor in the pace of change and uncertainty among HE teaching staff and management [5]. This is potentially compounded by the misconception that decolonisation entails censoring important Western content and replacing it with non-European content (promoting a so-called 'cancel culture'), while it truly is about adding context rather than removing content [1, 9]. Other misconceptions may relate to the concept of 'colour-blindness' as a form of meritocracy associated with the conviction that if everyone is treated equally, everyone will have equal chances [10, 11, 23, 24]. In other words, the idea that if teaching staff simply provide the same teaching and level of support to all students, irrespective of their skin colour, ethnicity and cultural background, then students will have equal opportunity to achieve. This is inadequate because it does not recognise the realities of structural and societal inequalities, birth advantage and disadvantage, and the cumulative impact of these upon students' lived experiences [25]. Furthermore, this conviction ignores issues of additional learning tax where minoritised students may need to invest more energy and labour than more privileged counterparts in order to experience physical and emotional safety, belonging and wellbeing, before attention can be focused on education [26]. Hence, different students will need different forms and levels of support throughout their degrees, in response to inequalities embedded within society and educational systems [7, 11, 23]. Furthermore, embracing 'colour-blindness' in HE disregards important aspects of students' identities [10].

## Brief historical background to the need for decolonising science

Britain began colonising North America in the sixteenth century, and continued expanding its empire around the globe for four hundred years. Concurrently, several other European empires, including the French, Dutch, Portuguese and Spanish each colonised major parts of the world, in particular the Americas and the Global South. The British and other European empires formally ended in the second half of the twentieth century, although a number of Western powers including the UK and United States continue to have small imperial holdings [27–29]. At this point power-dynamics in favour of the Global North were firmly established, not only with regards to political and socio-economic imbalances, but also in terms of science [5, 27]. Throughout our colonial history, scientific advances were continuously made as a direct result of the exploitation of imperial colonies, with credit given only to the White, male authors of scientific discoveries [5, 12, 13]. Sometimes these 'discoveries' were based on indigenous horticultural traditions, folk knowledge, and agricultural and medical practises in the colonies [27]. Although scientific advances were implemented in imperial colonies, non-Europeans were actively discouraged from gaining scientific and technological expertise. This meant that 'White' settler colonies, such as those in Australia and north America, were able to obtain scientific independence from Britain much earlier than other colonies, such as those in India and tropical Africa [27].

During the eighteenth and nineteenth centuries many scientists and scholars aimed to categorise humans into distinct 'races' according to physical characteristics, linked these differences to perceived intellectual capabilities, and ranked them. For example, Carl Linnaeus, the eighteenth-century Swedish botanist known as the 'father of modern taxonomy' divided human races into a hierarchy of sub-species with White Europeans at the top, and then red Americans, yellow Asians and black Africans in descending order below [27, 30, 31]. These hierarchies of human 'races' were used to justify the colonial enterprise already ongoing. This included the lucrative trade in enslaved human beings [32, 33], genocides, exploitations, and expulsions of perceived 'inferior' colonised populations, confirming the 'superiority' of White Europeans [27, 30]. The division of humanity into such qualitatively distinct 'races' were only abandoned by scientists in the late twentieth century as we have come to understand that the majority of human variation is cultural and clinal, rather than genetically and morphologically subdivisional [34].

Charles Darwin's development of the theory of evolution by natural selection, published in 1859, was a direct outcome of colonial actions. Travelling aboard the HMS Beagle, Darwin witnessed mass-killings of indigenous, colonised people first-hand. While Darwin struggled with the ethics of these actions he adhered to existing racialised views of colonised peoples [30, 35]. Later, using Darwin's published theories, his cousin Sir Francis Galton argued that human evolution could be 'sped up' by selective breeding to 'improve the human stock' [36]. In 1883, Galton—who laid the foundations for correlational statistics and regression analyses, and pioneered a range of methods in psychology—coined the term 'eugenics' [36, 37]. Eugenics was concerned with improving the human species by selectively breeding individuals with 'superior' physical and mental traits, while preventing 'inferior' individuals from reproducing [38]. 'The Francis Galton Laboratory for the Study of National Eugenics', established in 1907 at University College London, was directed by Karl Pearson—the inventor of the p-value and the Chi-Squared test—until 1933 [36]. Also in 1907, the Eugenics Education Society was formed in London, to 'further eugenics teaching at home, in the school, and elsewhere'. Membership of this society was popular until at least 1920, with members including 29 university staff and six politicians [36]. In 1925 Pearson created the scientific journal 'Annals of Eugenics' which still exists and publishes today, albeit under the more palatable name 'Annals of Human Genetics' [39].

The eugenics socio-political movement, mainstream in science education and research in Western societies throughout the first half of the twentieth century [36, 40], had an explicit focus on qualitative differences among human beings. This idea was linked with a claim that physical, mental and moral characteristics of people were overwhelmingly hereditary rather than determined by environmental factors. Embedded was the conviction that people classed as being 'feeble-minded', criminal, 'constitutionally weak', blind or deaf should be prevented from breeding [41]. These ideas were strongly supported by Sir Winston Churchill, but were never enacted into public policy in the UK [42]. However, they did form the basis of a number of government policies in the United States resulting in the forced sterilisations of some 60,000 people, performed mainly on poor, often African-American people in mental hospitals, before being condemned in 1936 [40, 43]. Scientific research conducted in the US and UK in the area of eugenics was an inspiration to Hitler [36], and even after World War II, it took some time for eugenics to be fully abandoned. For example, University College London continued to have a Chair of Eugenics, endowed by Galton, until 1963, the Eugenics Society changed its name to the Galton Institute only in 1989, and UCL renamed its Galton Lecture, Pearson Building and Pearson Lecture Theatre in 2020 after a public inquiry into their historical connections with the eugenics movement [41, 44].

Hence, for hundreds of years European empires colonised and exploited indigenous populations around the globe, claimed scientific discoveries based on findings from colonised lands and monopolised scientific and technological advances in the West at the expense of colonies and former colonies [45]. Education, knowledge and science was reserved–until extremely recently–for White males only [4, 12, 27]. While the first Black students–slowly–started obtaining university degrees in the early/mid nineteenth century in the UK [46, 47], women were awarded their first degrees from the University of London only in 1878, and from Oxford University and Cambridge University only from 1920 and 1948 respectively [48]. Even as formal European empires were ending in the second half of the twentieth century, the paradigm of qualitative differentiation of human beings continued to place White, wealthy men at the top of social, political and economic hierarchies as inherently 'better' human beings [49, 50]. Despite the widespread condemnation of the atrocities committed by Hitler and his Nazi Party based on their beliefs of racial superiority and race hygiene, the idea of 'race' and racial hierarchies has persisted.

We now know that environmental factors are strong determinants of intelligence and health, and while heritable genetic factors play some role in both, human populations around the globe differ genetically more *within* populations than *among* populations [51–54]. However, harmful, racial stereotypes from the colonial age are unfortunately still prevalent today, and are affecting students' learning in HE [17, 55, 56]. For example, to justify slavery, Black people were once proposed to have thicker skin and a much higher pain tolerance than White people. This fallacy still exists in medical education today, causing students to underestimate pain in Black patients and administer less pain relief compared to White patients [57]. Furthermore, despite the work of 'progressive biologists' in the UK, and within UNESCO on the international stage, in challenging ideas about races and 'racial inferiority', these ideas continue to hold sway within both science and public discourse [58]. Historians, and other scholars, continue to debate exactly why this idea endures with many arguing that its persistence is linked to its foundational role in our national and international political and economic systems [59, 60].

The result is an educational system and a science curriculum which is Eurocentric and almost exclusively celebrates scientific discoveries by White males [4, 9]. Teaching staff in science departments at UK universities are overwhelmingly White, and leadership roles are dominated by White men [61]. This means that Black, Asian and other students from marginalised

and minoritized demographics tend to experience a university where they are unlikely to see themselves represented among teaching staff, among scientific research staff, and among the scholars being celebrated in their curriculum. This is particularly true for female students from these demographics, or individuals in other ways intersecting multiple marginalised identities [6, 12, 13]. Such student experiences creates a barrier to feelings of belonging, and promotes a sense of exclusion [11, 13]. In this way, teaching practises, HE pedagogies and university structures reproduce racial inequalities and White privilege and entitlement, at a systemic level [13, 26]. This is reflected in marked awarding gaps (also sometimes referred to as attainment gaps) between White students and other ethnic minorities, with the highest gap in the likelihood of obtaining good degree outcomes being between White and Black students [62]. While there are a number of other important ways in which students may be disadvantaged, for example along the lines of socio-economic background, ability, gender identity, sexuality or neuro-processing, these are outside the scope of focus of the current study. While many of these inequalities are also the result of legacies of empire, this particular study examines the imperial legacies of racial thinking and racial inequalities. We have also used the term 'White' with a capital letter to denote that we are not talking about the individual activities or culpability of those with low melanin levels, but, instead, the functioning of Whiteness as a system of power and how it functions within science and HE [63, 64].

## Materials and methods

### Study population

This case study was conducted at a medium-large sized UK university (25–30,000 students) in the top 35 of the country's largest HE providers (out of 285 institutions) that attracts students from across the UK, and abroad, with the majority of students coming from Hampshire, Greater London, West Sussex, Surrey and Kent [65]. As in most UK institutions, the majority of academic staff members are White (at least 73.5% and likely more, with 12.6% of ethnicities unknown in 2021/2022), with about 13.8% identifying as Black, Asian, Mixed or 'Other' [65]. As in other UK institutions, staff members at this university come from around the UK as well as various European countries, with some staff members' heritage originating in other Western as well as non-Western countries (authors' personal observations). Hence, the observations from this case study are likely to be relevant to—and might even be representative of—HE institutions in the UK more broadly, although it was beyond the scope of the current study to evidence broader trends across the UK and beyond.

A questionnaire was constructed in Google Forms and distributed via email to all academic teaching staff in four science-related schools, covering the fields of Biological, Earth and Environmental Sciences, Biochemistry, Psychology, Pharmacy, Biomedicine, Geography and Geosciences, at the study university in March 2022. These four school were selected as the focus of this study was on science teaching. The remaining science-related schools at the institution had a more health and medical focus and were not included. This is because healthcare and medicine have a suite of additional challenges relating to the legacies of colonialism and racism due to patient-practitioner interactions [57, 66–70].

Reminder emails were sent out after three weeks and the survey was closed in June 2022 having achieved N = 46 submissions out of a total pool of 177 academic staff members in these four schools (i.e. representing 26% of science teaching staff).

### Ethical approval

Ethical approval was obtained in line with the University of Portsmouth Ethical Approval Processes (approval code: ED182319). The study was conducted according to the principles

expressed in the Declaration of Helsinki. Informed consent was obtained from all participants prior to data collection, responses were anonymous, and participants could withdraw consent up until submitting their response.

## The questionnaire

The questionnaire was comprised of five separate sections as described below (see SM1 for full questionnaire).

**Section 1: Consent.**    Participants were required to give consent for data sharing, and information was given about anonymity, along with an emotional trigger warning, and details about employee helplines.

**Section 2: Participant demographics.**    Respondents were asked to indicate their age, gender identity, ethnicity (according to [71]), number of years they had taught in HE, and to which degrees they contributed teaching. If an approximate age group was submitted we allocated an age estimate for data analyses (i.e. '30s' became 35, '40–50' became 45, '50–55' became 53, and '>60' became 65; N = 10). Similarly, we allocated an estimated number of years in HE ('<1' became 1, '>20' became 22, and '>25' became 27; N = 3). This data is summarised in the results section, but the raw data is left out of our openly available dataset to ensure full anonymity of participants.

**Section 3a: Familiarity with and understanding of decolonisation.**    This section aimed to evaluate participants' 1) self-assessed understanding of decolonisation; 2) self-assessed familiarity with actions to decolonise teaching content; and 3) level of agreement with specific concepts as presented in key literature; and 4) misconceptions.

To assess (1) and (2) we asked participants to first rate how well they understood the concept of decolonising the curriculum in HE from 1 (I do not know what it means) to 5 (I have a full understanding), and next, how familiar they were with actions they can take in their teaching to decolonise their own teaching content and pedagogical practises on a scale from 1 (I'd have no idea where to even begin) to 5 (I know how to decolonise my teaching).

To assess (3) and (4), we presented six statements and asked respondents to indicate to what extent they agreed with each statement on a scale from 1 (disagree) to 5 (fully agree). They were asked to skip if unsure, although a response of '3' was also taken to indicate uncertainty. Four of the statements represented common descriptions of decolonisation in the literature [4, 5, 9, 11, 13, 23, 72] while two represented misconceptions about decolonisation [5, 9, 11, 23] (SM1).

**Section 3b: Benefits and risks.**    To assess participants' perception of the potential benefits as well as downsides or risks associated with decolonising the science curriculum, we listed ten suggested benefits and risks and asked participants to tick all they agreed with or believed apply. These suggestions were based partly on suggestions from the literature [9, 13] and partly based on downsides and risks as suggested by colleagues of LG in private conversations (SM1). Participants could list additional benefits and risks in an 'Other' free-text option.

**Section 3c: Topics of training.**    To assess participants' perception of training needs, they were asked which of a list of six topics of training they thought themselves, their colleagues, and/or management/leadership would benefit from training in.

**Section 4: Teaching activities.**    To assess the current state of decolonisation of the science curriculum at the institution we listed two sets of suggested actions/activities that teaching staff can do to start decolonising their teaching content and delivery, inspired by the University of Sheffield's toolkit for Contextualising the Curriculum in Ecology and Evolutionary Biology [72]. These two sets differed by being actions that can be done either *with* (6 actions) or *without* necessarily talking to students directly about topics relating to decolonisation (5

actions). For example, actions *without* direct discussion included diversifying the reading list and including non-Western examples, while actions that *require* direct discussions included e.g. talking to students about ethnicity, race and racism. For each activity, participants were asked to indicate what best applied to them out of: 'I tend to do this', 'I am prepared to do this now', 'I would like to do this, but I need training first', 'I would not feel comfortable doing this', 'I do not feel this is relevant or needed in my teaching', and 'Unsure'.

**Section 5: Importance, responsibility and barriers.** In the final section we asked participants to rate how important they felt it is to decolonise the science curriculum at their institution from 1 (Not important) to 5 (Extremely important). We then asked whose responsibility it is to implement decolonising efforts: Teaching staff, The EDI committee, A new committee dedicated to decolonisation, and/or School, Faculty, and/or Institution management. Finally, we asked which barriers there might be to respondents decolonising their own teaching content. We listed 22 potential barriers based on suggestions from the literature as well as suggestions derived from conversations with colleagues (SM1).

**Optional free text in sections 2 to 5.** At the end of sections 2 to 5 respondents had the option of writing free text. It is beyond the scope of this paper to analyse this qualitative data. A full qualitative analysis of this data will be published in a subsequent paper.

## Data analyses

All statistical analyses were done using the software R, version 4.1.1 [73].

**Pairwise comparisons of teaching actions.** In the questionnaire we had suggested a set of six teaching activities *without* directly talking to students about diversity and inequality, and a set of five activities that *included* direct discussions with students. For each participant and for each response option ('tending to', 'being prepared to now', etc) we calculated a likelihood (from 0 to 1) of e.g. 'tending to do' activities both *with* and *without* directly addressing the topic by averaging the binary responses across each set of activities. We then compared the likelihood of 'tending to', 'being prepared to' etc. of participants to engage in activities *with* and *without* directly discussing the topic with students using non-parametric Wilcoxon tests for matched pairs.

**Generalised linear models for testing effects of demography.** We constructed three generalised linear models (glms) to test for potential effects of gender, discipline and number of years in HE on each of three response variables: self-assessed understanding of decolonisation, familiarity with teaching activities, and importance rating of decolonisation. We had aimed to test for potential effects of ethnicity as well, but a low diversity in ethnicities of our participants prevented this approach.

All models were fitted with a Poisson error distribution and log link function due to response variables being integers. We confirmed the absence of overdispersion using a goodness of fit test before proceeding with significance testing. The relatively low sample size (N = 42 resulting from two missing values in discipline and two in gender) prevented us from testing for potential interaction effects among predictor variables. Number of years in HE was included instead of age to avoid further reduction in sample size from missing values in age (justified by a strong, positive Spearman's Rank Correlation between age and year in HE: rho = 0.79, $p < 0.001$, S1 Fig in S2 Appendix).

## Results

### Respondent demographics

Out of 46 responses, 23 indicated a female gender identity (50.0%) and 21 indicated male (45.7%). Zero respondents indicated non-binary and two (4.3%) did not indicate a gender and were excluded from analyses that included gender.

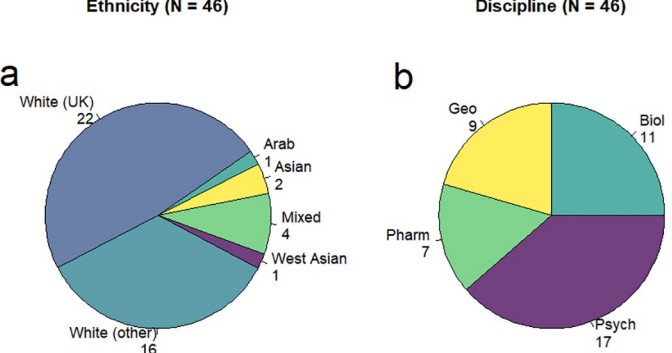

**Fig 1. Ethnicity and discipline.** Pie charts of a) ethnicity and b) discipline of the respondents. 'Geo' includes participants mainly contributing to degrees in geology, geography, palaeontology and environmental sciences. 'Biol' includes biology, marine biology, and biochemistry. 'Pharm' includes pharmacy, biomedical sciences, pharmacology and medical biotechnology, while 'Psych' includes various psychology degrees.

Age of respondents ranged from 33 to 71 years, with a median of 45 (mean = 46.2; st. dev = 10.0). One participant did not indicate an age. Respondents had taught in HE from <1 to 40 years with a median of 13 (mean = 14.6; st.dev = 9.1).

The ethnicity of respondents was overwhelmingly White (82.6%) with 22 respondents (47.8%) being White with a UK background and 16 (34.8%) non-UK White (Fig 1a). Zero respondents identified as Black. Four respondents (8.7%) indicated Mixed or Multiple ethnicities while another four (8.7%) identified as Asian or Asian British, or self-described in the 'Other' option as West Asian or Arab (Fig 1a). While a more diverse sample would have been preferable, the (low) diversity in ethnicities does closely resemble the actual constitution of academic teaching staff at this institution [65] and may therefore be considered representative of the study population, and fairly representative of UK institutions more broadly [65].

Respondents contributed to a wide range of scientific BSc and MSc degrees (Fig 1b). The most represented degrees were BScs in Psychology (16 respondents), Biology (12), Marine Biology (9), Biochemistry (9), Biomedical Science (6), Marine Environmental Science (6), Environmental Science (5), and Geography (5) (S2 Fig in S2 Appendix).

## Familiarity, misconceptions and importance

Self-assessed understanding of decolonisation showed a high median of 4 (mean = 3.44; Fig 2a), although 9 participants (19.6%) rated their understanding only 1 or 2. Familiarity with

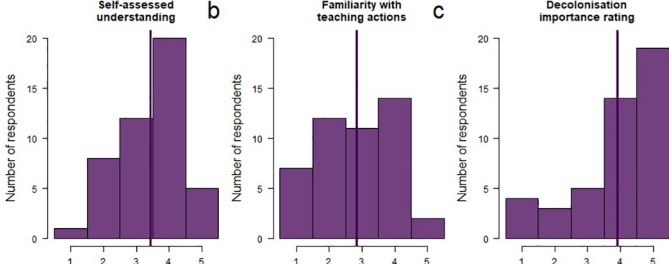

**Fig 2. Understanding, familiarity with teaching actions, and importance.** Histograms of responses from 1 (low) to 5 (high) of participants' a) self-assessed understanding of decolonisation, b) self-rated familiarity with teaching actions towards decolonising their content and delivery, and c) rating of how important they felt it is to decolonise the science curriculum at their institution. Vertical lines indicate mean values.

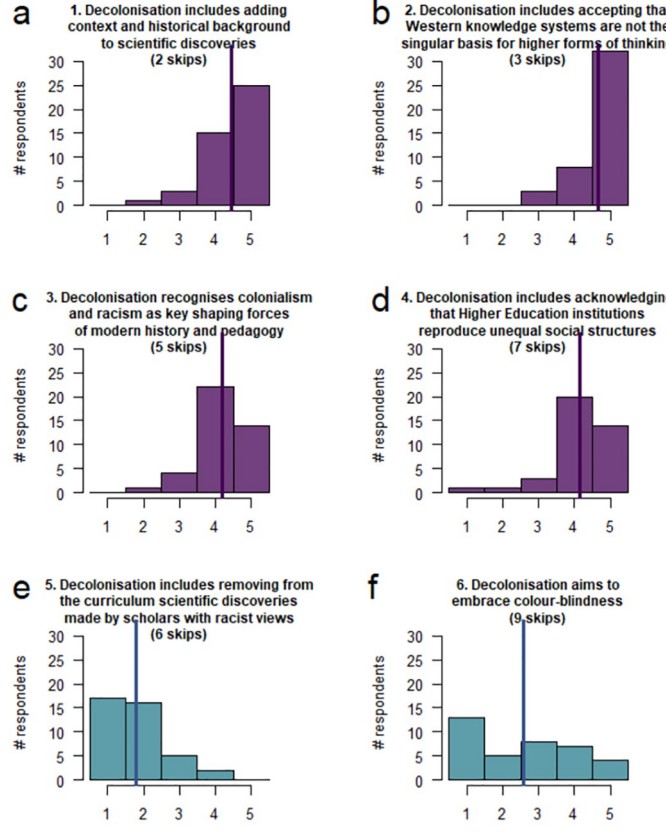

**Fig 3. Agreement with statements about decolonisation.** Histograms of the distribution of agreement ratings from 1 (disagree) to 5 (agree) of statements #1–4 (a-d; purple colour) reflecting decolonisation as described in the literature: 1 [13, 72], 2 [11, 13], 3 [4, 13] and 4 [9, 13], and statements #5–6 (e-f; blue-green colour) reflecting misconceptions: 5 [5, 9] and 6 [11, 23]. Vertical lines indicate mean values. Statements were presented in the following randomised order in the survey as not to identify which were misconceptions: 3, 4, 5, 1, 6, 2.

teaching actions scored lower with a median of 3 (mean = 2.83; Fig 2b), and 19 participants (41.3%) rating this 1 or 2. Participants generally rated the importance of decolonisation high, with a median of 4 (mean of 3.91; Fig 2c), although 7 participants (15.2%) rated the importance low, 1 or 2.

Participants generally showed agreement with the four statements about decolonisation that reflect how decolonisation is described in the literature (statements #1–4; Fig 3a–3d) with means ranging from 4.2 to 4.7 and only one or two participants outright disagreeing with each statement (responding '1' or '2', Fig 3a–3d).

The number of participants that rated a question 3 or lower or skipped one of the four questions (i.e. were unsure or disagreed) were as follows: statement #1: 6 (13.0%), #2: 6 (13.0%), #3: 10 (21.7%) and #4: 12 (26.1%) indicating relatively high uncertainty about HE reproducing unequal social structures.

The misconception that decolonisation includes removal of scientific discoveries made by scholars with racist views received general disagreement with a mean of 1.8, while 13 participants (28.3%) scored this 3 or higher, or skipped it (Fig 3e).

The misconception that decolonisation aims to embrace colour-blindness was skipped by the highest number of participants (9 skips) and produced the most varied responses with a mean of 2.6 and 28 participants (60.9%) scoring 3 or higher, or skipping it (Fig 3f). This

indicates a high level of uncertainty and misconceptions about colour-blind behaviour in relation to decolonisation.

Four participants had skipped three or more of the six statements about decolonisation. These four participants were exclusively >60 years of age and White (three UK and one non-UK), three were male and one female.

We tested the effects of age, gender, and discipline on each of the three response variables self-assessed understanding, familiarity with teaching actions, and importance of decolonisation. All three response variables generally decreased with age (S3 Fig in S2 Appendix); the medians for both self-assessed understanding and importance of decolonisation were lower for males than females (3 versus 4, and 4 versus 5 respectively, S4 Fig in S2 Appendix); and Psychology and Geography/Geology showed slightly higher medians in understanding of, teaching familiarity and importance as compared to Biological Sciences and Pharmacy/Biomed (S5 Fig in S2 Appendix). However, all of these effects were statistically not significant as all three full models were non-significant (all full glms: p > 0.4).

## Benefits and risks

The majority of participants acknowledged several suggested benefits of decolonisation. Out of 46 participants, 41 (89.1%) agreed that decolonisation ensures that students of all ethnicities see themselves in their education and feel some level of ownership over the knowledge they learn. Four other benefits were acknowledged by a majority of 31–38 participants (67.4–82.6%; Fig 4).

However, a notable minority of participants felt that decolonisation was associated with risks (Fig 4), in particular the risk of promoting a 'cancel culture' (11 participants, 23.9%), and a fear that addressing colonialism, racism and oppression risks creating more division and disputes rather than promoting inclusion and belonging (6 participants, 13.0%).

| Type | Statement | Number of participants |
|---|---|---|
| Benefit | It ensures that students of all ethnicities see themselves in their education and feel some level of ownership over the knowledge they learn | 41 |
| Benefit | It prepares students of all ethnicities for a modern, multi-cultural world | 38 |
| Benefit | It helps to reduce inequalities, privilege and entitlement in society as a whole | 35 |
| Benefit | It helps to break the cycle of racial inequalities currently being reproduced by Higher Education | 32 |
| Benefit | It would attract and retain more Black, Asian and other minoritized students, helping to close the awarding gap | 31 |
| Risk | It risks promoting 'cancel culture' where important traditional content is being censored out | 11 |
| Risk | Addressing the role of colonialism, racism and oppression in the history of science risks creating more division and disputes rather than promoting inclusion and belonging | 6 |
| Risk | Including historical contexts means having to remove some of the content currently taught, risking vital topic details being removed from the curriculum | 5 |
| Benefit/misconception | There is no real benefit to White students, but it will benefit Black, Asian and other minoritized students, increasing their feeling of belonging and representation | 2 |
| Risk | It risks making our department appear too radical, as the term is linked with controversy and activism | 1 |

**Fig 4. Benefits and risks.** Suggested benefits and risks ranked and colour-coded according to the number of participants agreeing with each (purple = high to yellow = low).

Nine participants used the 'Other' option to suggest additional risks, benefits, and comments (S1 Table in S2 Appendix).

## Training needs

A majority of participants felt that training would be beneficial for both themselves, their colleagues, and for management/leadership in all suggested topics of training (Fig 5). The highest ranked training needs for oneself and colleagues were actions that can be done in our teaching (36 and 34 participants respectively, 78.3% and 73.9%) whilst the highest ranked training need for management/leadership was what can be done at a structural level (34, 73.9%). For each of the six topics of training suggested, 4–6 participants (8.7–13.0%) did not think training was needed for anyone (Fig 5).

## Teaching actions

Many participants indicated that they currently tend to do some of the suggested activities that do not include direct discussion (Fig 6a), in particular to include more global and diverse examples (30 participants, 65.2%) and include the historical development and context of the topic (26, 56.5%). Just fewer than half of participants (19–22, 41.3–47.8%) diversify their reading list and lectures to include contributions and perspectives from women and scholars that are not White (Fig 6a) and many respondents were either prepared to do these activities now or would like to do them after receiving training (15–20, 32.6–43.5%).

Significantly fewer participants tended to do actions that include discussions with students as compared to actions without discussions (Wilcoxon tests for matched pairs, $V = 577.5$, $p < 0.001$, Fig 6a vs 4b, S6a Fig in S2 Appendix). Indeed, only 8 participants (17.4%) talk to students about decolonisation, although a further 23 (50.0%) were either prepared to do this now or would like to do so after receiving training (Fig 6b). A minority (14–19, 30.4–41.3%) talk to students about ethnicity, race, racism, the over-representation of White scholars in science, or discuss the influence of colonialism on the understanding of their topic, or who/who is not acknowledged for scientific discoveries (Fig 6b).

Although only 0–4 participants (up to 8.7%) felt uncomfortable with each suggested teaching activity, significantly more participants felt uncomfortable with actions *with* versus *without* direct discussions (Wilcoxon tests for matched pairs: $V = 3$, $p = 0.014$, S6e Fig in S2 Appendix). Indeed, participants were generally more positive about topics *without* (Fig 6a) versus *with* (Fig 6b) direct discussion (tending to, prepared to now or after training pooled, $V = 244$, $p = 0.0013$, S6d Fig in S2 Appendix).

| Topics for training | Parties that would benefit from training | | | | |
| --- | --- | --- | --- | --- | --- |
| | Yourself | Your colleagues | Management Leadership | Not needed | Unsure |
| What can be done by teaching staff (i.e. actions you can take in your own teaching)? | 36 | 34 | 27 | 4 | 4 |
| What are the aims of decolonisation (i.e. what are we trying to achieve / what are the success criteria)? | 35 | 33 | 30 | 5 | 4 |
| The historical background to current racial inequalities in Higher Education and Science (i.e. what has led to the current state)? | 32 | 30 | 28 | 6 | 4 |
| What is decolonisation? | 30 | 30 | 28 | 5 | 4 |
| Why is decolonisation needed in Higher Education today (i.e. what is the current state) | 28 | 28 | 27 | 6 | 6 |
| What can be done at a structural level (i.e. actions required by management/leadership)? | 27 | 24 | 34 | 5 | 2 |

**Fig 5. Training.** Suggested topics for training and the number of participants (color-coded purple = high to yellow = low) that thought training within each topic would be beneficial for various parties.

**Fig 6. Teaching.** Suggested teaching activities a) *without* and b) *with* talking to students directly about diversity and inequality, and the number of participants (color-coded purple = high to yellow = low) indicating their preparedness to implement each activity.

Participants were much more likely to already tend to do activities *without* direct discussion (V = 577.5, p < 0.001, S6a Fig in S2 Appendix) and more likely to require training before doing activities *with* direct discussion (V = 38, p = 0.013, S6c Fig in S2 Appendix), while there was no difference between activities with and without discussion in whether participants were prepared to do them now (V = 299, p = 0.17, S6b Fig in S2 Appendix) or whether they find them not relevant in their teaching (V = 23.5, p = 0.073, S6f Fig in S2 Appendix).

For each suggested teaching activity between 4 and 9 participants (8.7–19.6%) felt the activity was not needed in their teaching (Fig 6). For each action requiring direct discussions a minimum of 7 participants (15.2%) felt it was not needed in their teaching.

## Responsibilities

A majority of participants felt it was the responsibility of teaching staff to decolonise the curriculum (38 participants, 82.6%, Fig 7). Between 30 and 35 (65.2–76.1%) also felt that management at School, Institution and Faculty levels bears responsibility. Four participants (8.7%) did not think decolonisation is needed at their institution. Four participants used the 'Other' option to add additional comments (S2 Table in S2 Appendix).

| Suggested responsibilities | Number of participants |
|---|---|
| Teaching staff | 38 |
| School management | 35 |
| Institution management | 33 |
| Faculty management | 30 |
| The EDI (Equality, Diversity and Inclusion) committee | 27 |
| A new committee dedicated to Decolonisation | 5 |
| We do not need to implement decolonisation efforts at [this institution] | 4 |

**Fig 7. Responsibility.** The number of participants (ranked and color-coded purple = high to yellow = low) indicating whose responsibility it is to decolonise the curriculum.

## Barriers

The most commonly indicated barriers for teaching staff to decolonise their curriculum was a lack of training in the topic in general, not having enough time to do it properly, and not having a full understanding of the problem (15–19 participants, 32.6–41.3%, Fig 8).

| Suggested barriers | Number of participants |
|---|---|
| I lack training in the topic in general | 19 |
| I don't have the time needed to do this properly | 18 |
| I don't have a full understanding of the problem | 15 |
| Terminology in this topic is confusing and I often don't know which terms to use | 14 |
| I don't know who to get ideas and advice from | 13 |
| I'm afraid of offending someone | 11 |
| I don't have enough support from leadership/management | 11 |
| EDI and decolonisation efforts are not recognised in my workload plan | 9 |
| Structural changes are needed in my school/institution before I will engage with this | 6 |
| I don't know which actions I can take in my teaching | 6 |
| Decolonisation should be taught in dedicated sessions in e.g. skills-related modules rather than be implemented in most modules | 6 |
| My modules are packed and there's no room to add historical context to the topic without having to remove important content | 5 |
| I think it's less important than other more pressing EDI matters | 4 |
| Instead of me decolonising my teaching, we should get external experts in to teach decolonisation to our students | 4 |
| I am non-White and feel exhausted from always having to be the one addressing these matters | 3 |
| Decolonisation is only relevant for subjects such as History and not for Science | 2 |
| I don't agree with the concept of decolonisation | 1 |
| I have tried some decolonisation efforts but it was not received well by my students | 1 |
| The topic makes me uncomfortable | 0 |
| All, or most, of my students are White so there is no real need to decolonise my content | 0 |
| I am White and therefore do not feel it's appropriate for me to talk about this topic | 0 |
| It is beyond the scope of my job to become trained in this topic | 0 |

**Fig 8. Barriers.** Suggestions of barriers preventing participants from decolonising their curriculum. Number of participants choosing each suggested barrier is ranked and colour-coded (purple = high to yellow = low).

Importantly, three out of the eight respondents who identified as of other ethnic origin than White (37.5%) also identified 'exhaustion from always having to be one addressing these matters' as a barrier.

Zero respondents chose as barriers that decolonisation was beyond the scope of their jobs or that them or their students being White made it inappropriate or unnecessary for them to engage in decolonisation. Nine participants used the 'Other' option to add additional comments (S3 Table in S2 Appendix).

## Discussion

We explored science teaching staff's perceptions of and engagement with decolonising the science curriculum at a HE UK institution and found that dispositions were overwhelmingly positive. Participants in this case study tended to rate the importance of decolonisation highly but rate their understanding of the concept lower. The vast majority (80–90% of participants) agreed that decolonisation can benefit students of all ethnicities and felt that they, as teaching staff, had a responsibility to decolonise the curriculum. Furthermore, nearly half of participants already tend to take some steps in their teaching to diversity their reading lists and teaching materials, and more than ¾ of participants were positive about decolonising their teaching. However, at least one in five indicated they need training before they feel comfortable taking further steps in their teaching, in particular regarding discussing sensitive topics with students like the influence of colonial actions and racism on their scientific topic. Hence, we found that there is motivation and drive towards creating a more inclusive and diverse curriculum among a majority of scientists teaching and training future generations of scientific researchers in our UK HE case study. However, we also identified considerable barriers as well as critical, common misconceptions that may hinder teaching staff's engagement with decolonisation. These misconceptions and barriers are likely to slow the process of creating more equal and just teaching and learning environments in UK HE science degrees that feels welcoming and accessible to students of all cultures and backgrounds [3, 6, 9, 15, 56, 74].

One of the top most identified barriers to decolonising one's own teaching content was not having a full understanding of the scope and breadth of decolonising HE, and almost one in five of our participants rated their own understanding of decolonisation low. This limited understanding was further evident in the identified misconceptions: almost ¼ of participants agreed with, or were unsure about, the misconception that decolonisation includes removing scientific discoveries made by scholars with racist views, and a similar fraction feared that decolonisation promotes a 'cancel culture' where important, traditional content is censored out. Unfortunately, some UK media outlets spread and reinforce these misconceptions reporting on steps towards 'censoring out' seminal White scholars in academia, even though decolonisation activities in reality call for added historical context of the content and a greater representation of minoritised scholars [5, 8]. The literature on decolonisation clearly states that radical change in HE is needed, not in the form of erasing current content, but rather by openly discussing the often-gaping void of content produced by non-Western scholars [1, 4, 5, 9, 13]. This spreading of misinformation about the decolonising agenda by the media may shape, and negatively affect, the attitudes of HE teaching staff towards the very idea of 'decolonising' [8]. A separate question that might be fruitful to discuss is whether similar approaches would meet less resistance if rephrased as 'contextualising' the curriculum [72] or improving equity, diversity and inclusion in a science teaching and learning context [74]. However, here we maintain the decolonisation terminology due to the concept now being firmly established and defined in the literature in relation to pedagogy, HE teaching and learning, research and scientific practises [1, 2, 4, 8, 9, 11–13, 15, 16, 75–80].

Decolonisation requires addressing the influence of racist and oppressive historical paradigms on the direction of scientific research and interpretations of scientific findings. This requires honest, open and historically informed discussions about race, gender, positionality, intersectionality, privilege and marginalisation [3, 9, 10]. However, in our study we found that about 60% of participants agreed with, or were unsure about the misconception that decolonisation aims to embrace colour-blindness. A colour-blind framework aims at treating everyone exactly the same, irrespective of individual cultural backgrounds, lived experiences, skin colour and ethnicity [24]. This approach is problematic because it disregards historical, systematic and societal differential treatments of people of different demographics and the way this affects the science curriculum, the university experience, and the learning environment in HE. Such a disregard for important aspects of students' identities and the impact of structural inequalities on their lived experiences would prevent the much needed open and honest discussions about the links between racism, colonialism and science and HE [4, 10, 11].

Despite the motivation to decolonise the curriculum expressed by a majority of our participants, we found that many were unsure how to do so at a practical level. Forty percent rated their familiarity with teaching actions low, and more than ¾ felt they, and their colleagues, would benefit from training in what can be done by teaching staff. Indeed, many different steps can be taken by teaching staff in HE to start the journey towards decolonising their content and delivery [11, 72], and many staff members may already be teaching in ways consistent with the decolonisation agenda. For example, about 60% of our participants already tend to include the historical development and context of their topic, include pictures of celebrated scholars in PowerPoint slides, and include more global and diverse examples. These types of activities, if accompanied by a discussion of the people(s) that are not appropriately accredited for the discoveries, and the historic reasons why, are important first steps in the journey towards decolonising the science curriculum [1, 11, 72]. However, participants were significantly less likely to take that extra step of talking directly with students about topics such as the overrepresentation of scientific contributions from White people, who is not acknowledged for the great discoveries, and how colonial actions and racism has influenced the current understanding of the topic. Participants also felt more uncomfortable about discussing these issues with students as compared to taking actions that did not include direct discussions. Hence, upskilling staff not only with tools, practical ideas, and relevant examples of how to decolonise their content, but also equipping them with the necessary literacy on the subject, and the confidence to have important conversations with students is crucial for a successfully decolonised science curriculum and learning space [1, 9, 11, 13]. Such upskilling could take the form of staff training sessions tailormade for discussing inclusive teaching and pastoral care practises, with a focus on appropriate terminology [74], in the context of the historical development of biological sciences embedded within colonial paradigms, social injustices and inequalities [1, 6].

A report by Higher Education Policy Institute that consulted both staff and students at UK HE institutions concluded that decolonisation is vital for both staff and student wellbeing and suggest that decolonisation efforts will not only help individuals and institutions, but is also key to a stronger democracy and society [9]. However, decolonisation requires addressing historical and current racial inequalities, and the process directly relates to 'colonisation' and its history of violence, racism and slavery [4, 5, 11]. These are emotive topics that will evoke complicated and unpredictable emotional responses in students depending on their ethnicity, background and lived experiences [18]. Hence, such conversations must be handled carefully and respectfully in a teaching and learning environment. The fear expressed by 13% of our participants that discussing such sensitive topics may create more division and disputes than inclusion and belonging, together with the feeling that terminology is confusing, and the fear

of offending someone, as expressed in about ¾ of participants, must be taken seriously. Having the appropriate vocabulary with which to discuss race, inequality and marginalisation with students, and having the ability to create a safe and open learning space is at least as important for teaching staff to fully decolonise their teaching as is a deeper understanding of the historical development and global context of their specific, scientific topic.

However, we may need to accept that not all teaching staff will ever feel capable of discussing these topics with students. In our case study, 15–20% of teaching staff did not find it relevant to discuss topics of race and (de)coloniality in their teaching, with a non-significant tendency for older staff and males to rate the importance of, and their own understanding of, decolonisation low. Case studies at other institutions suggest that under-developed or tokenistic decolonising efforts do not improve students' well-being and belonging [18, 19, 21]. Indeed, if efforts to diversify content does not fully acknowledge the variety of student identities, or if discussions are addressed with noticeable discomfort or a lack of recognition of one's own privilege, such efforts can cause further feelings of alienation in minoritized students [19]. Hence, it is crucial that management does not pressure teaching staff who feel unprepared, unmotivated or uncomfortable to open up emotive discussions with students, as this could lead to undesired outcomes. Decolonisation efforts should be encouraged, and–as discussed below–training must be provided to support this, and our findings suggest that a majority of staff members will engage with this positively, even if a minority of educators may be unwilling to engage. Studies of student experiences suggest that having one or a few lecturers or tutors that are open, willing to discuss emotive topics, and show they care greatly increases feelings of belonging–especially for minoritized student [19, 81]. This does not require all teaching staff to be equipped for emotive conversations and so efforts may be best placed upskilling and training the large proportion of staff members who have a keenness to engage.

Several universities have published helpful decolonisation toolkits that provide suggestions for HE teaching staff of actions and activities they can implement in their teaching and learning activities [11, 72]. However, our results strongly suggest that staff training, in conjunction with the use of toolkits, is needed to progress with decolonising the science curriculum in UK HE. The top most commonly identified barrier to decolonising the science curriculum was a lack of training in the topic in general. The absence of departmental training on decolonising the science curriculum was evident in the identified common misconceptions, in the uncertainty among participants about how to decolonise their content, and in the relative reluctance to discuss emotive topics with students. We found that up to ¾ of participants felt they, their colleagues, and management would benefit from training in what decolonisation means and what its aims are, why it is needed in HE, and the historical background to current racial inequalities in HE and science. Importantly, zero respondents felt it was beyond the scope of their job to become trained in decolonisation. However, not enough time to do it properly, and a lack of support from management were also common barriers. Several participants felt that structural changes were needed in their school or institution before they would engage with this and almost ¾ felt that management would benefit from training in what can be done at a structural level. However, some participants–in particular those who harboured misconceptions about decolonisation–were unwilling to receive training. Indeed, about 10% of participants felt that training was not needed on any topic related to decolonisation for any staff members, and that decolonisation was not needed at their institution. Hence, when training is provided, this is likely to be met by resistance from a small proportion of staff members. As argued above, efforts may be most optimally spent focussing on training the large proportion of staff members who are willing and motivated.

While science education in the UK was the focus of this study, we recognise that attitudes to, and progress with, decolonising the curriculum is likely to differ across disciplines, and in

other regions of the world. As the authors themselves enjoy academic careers that span the humanities, social and biological sciences, we recognise that knowledge is contextually legitimised and specialised within fields of practice by the actors within them, with varying epistemological and ontological interpretations [82]. This may result in disciplines having different understandings of the concept, practice and application of decolonising, with varying levels of commitment and motivation to de-centre western dominance. We expect that steps towards decolonising the curriculum are more developed in the humanities and social sciences, compared to the life sciences, with a deeper understanding of the concept and less resistance towards it. This is because the decolonisation movement at UK universities started within the humanities, and most research has been published within related fields [8, 83–85]. However, while conversations and practices related to decolonising may have been taking place within the humanities and social sciences longer than in the life sciences, they are not without controversy and there is still a lot of work to be done to ensure fully socially just and equitable education and curricula across disciplines in the UK and beyond. The specific requirements and challenges related to decolonising the curriculum might differ between disciplines, but we believe that the trends identified in this case study are illuminating and relevant across disciplines in the life sciences, social sciences and humanities.

## Conclusions

Our colonial legacy continues to impact our society and education, reproducing racial inequalities in UK HE. Students of minoritized ethnicity report feelings of isolation, exclusion and of not belonging [9–11]. Decolonising Universities, as one strand of global decolonisation, aims to rebuild the educational system in a more equitable, socially just and representative way. It does this by incorporating the historical context for White and Western over-representation of past and current scholars, benefitting students of all backgrounds and ethnicities [4, 9, 13]. This is a bold aim that will require many steps along the way. In our UK HE case study we find an encouragingly positive disposition of teaching staff towards decolonising the science curriculum, a thirst for a deeper understanding of what exactly decolonisation aims to achieve, and concrete tools and relevant examples that staff can use in their own teaching. However, misconceptions linking decolonisation to 'cancel culture', and barriers such as an apprehension to talk to students about race, and a genuine lack of time due to high workloads, are currently slowing curricular transformation.

Based on these findings, we propose that training is needed both at an institutional level, to educate leadership and management of the need to support staff to create teaching and learning spaces that are diverse and welcoming to all, and at departmental level for teaching staff. Training for science teaching staff needs to be tailored to the scientific topics taught and researched in a given department. For example, we found that some participants had the misconception that decolonisation was only relevant for subjects such as History and not for Science. Hence, historical examples of oppression, exploitation and marginalisation within specific fields of science must be included in decolonisation training to maximise the relevance of the training. Furthermore, training must include a set of concrete and subject-specific tools and activities that teaching staff can readily utilise, as the time staff have available for curriculum change is extremely limited (see helpful suggestions specific to the fields of ecology, evolution and conservation in Contextualising the Curriculum at the University of Sheffield [72]). Finally, there is a need for staff to develop their skills in creating safe spaces in their teaching to confidently discuss emotive topics, and for equipping them with the appropriate vocabulary for such discussions. Only when students experience teachers that are comfortable openly discussing topics of race and decolonisation will we be able to address inequality, and empower

all students with a feeling of belonging. Additionally, only when scientists that teach and train future generations of scientists become prepared to address the historical—and current—global biases and injustices in science production and access to knowledge will scientific practises become more inclusive and just.

## Supporting information

**S1 Appendix. Questionnaire.**
(DOCX)

**S2 Appendix. Supplemental figures and tables.**

- Supplementary figures

  - S1 Fig: Correlation between age and number of years in HE

  - S2 Fig: Degrees contributed to

  - S3-S5 Figs: Effects of age, gender and discipline on perceptions

  - S6 Fig: Pairwise comparisons of types of teaching activities

- Supplementary tables

  - S1 Table: Benefits and risks

  - S2 Table: Responsibilities

  - S3 Table: Barriers
    (DOCX)

**S3 Appendix. Raw data.**
(XLSX)

**S4 Appendix. R code.**
(R)

**S5 Appendix. Resumen científico Español (Abstract in Spanish and English).**
(DOCX)

## Acknowledgments

We would like to thank the Heads of Schools for permission to distribute the surveys and all the participants of this study who filled in the questionnaire. Thanks to AJ Fleming for translating the abstract into Spanish and to several colleagues for providing helpful comments on earlier versions of the manuscript. LG would like to thank J Brindley for continued support.

## Author Contributions

**Conceptualization:** Lena Grinsted.

**Data curation:** Lena Grinsted.

**Formal analysis:** Lena Grinsted.

**Investigation:** Lena Grinsted.

**Methodology:** Lena Grinsted.

**Visualization:** Lena Grinsted.

**Writing – original draft:** Lena Grinsted.

**Writing – review & editing:** Lena Grinsted, Catherine Murgatroyd, Jodi Burkett.

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
