## [Decision Letter · Decision Letter 0]

22 Jul 2024

PONE-D-24-23238Exploring attitudes to decolonising the science curriculum – a UK Higher Education case studyPLOS ONE

Dear Dr. Grinsted,

Thank you for submitting your manuscript to PLOS ONE. After careful consideration, we feel that it has merit but does not fully meet PLOS ONE’s publication criteria as it currently stands. Therefore, we invite you to submit a revised version of the manuscript that addresses the points raised during the review process.

I would recommend that paper needs minor revisions. The reviewer is clear in all those specific aspect that the article needs some work. Please, consider all those suggestions and include them in your article, as those will improve the content of this work. 

We look forward to receiving your revised manuscript.

Kind regards,

Francisca Ortiz Ruiz, Ph.D.

Academic Editor

PLOS ONE

Journal Requirements:

Additional Editor Comments:

Dear author,

Thank you very much for your submission, and the opportunity to read your research.

This paper needs minor revisions. The reviewer is clear in all those specific aspect that the article needs some work. Please, consider all those suggestions and include them in your article, as those will improve the content of this work.

Kind regards,

Reviewers' comments:

Reviewer's Responses to Questions

**Comments to the Author**

1. Is the manuscript technically sound, and do the data support the conclusions?

Reviewer #1: Yes

2. Has the statistical analysis been performed appropriately and rigorously? 

Reviewer #1: Yes

3. Have the authors made all data underlying the findings in their manuscript fully available?

Reviewer #1: Yes

4. Is the manuscript presented in an intelligible fashion and written in standard English?

Reviewer #1: Yes

5. Review Comments to the Author

Reviewer #1: The research analyzes the attitudes towards decolonising the science curriculum among academic teaching staff at a medium-large UK university. There is a lot to like about this work. Although it is not experimental, it presents a valuable case study using a rich dataset from a questionnaire covering demographic characteristics, understanding of decolonization, benefits and risks, training and teaching activities, and the importance and barriers to decolonising the science curriculum. The brief historical background section is also appreciated. Overall, I enjoyed reviewing this manuscript.

I find the manuscript to be a technically sound piece of scientific research, with data that support the conclusions and specific recommendations. My primary concern is the lack of explanation regarding the science-related schools under analysis (Biological Sciences, Psychology, Pharmacy, Biomedicine, Geography, and Geosciences). What criteria were used to select these four schools? The central argument of the manuscript is also applicable to social sciences, so an explanation of the selected fields would enhance the discussion and policy implications.

The statistical analysis is performed appropriately and rigorously. The methods are suitable given the small sample size. However, a deeper intersectional analysis could provide further insights into the heterogeneity of attitudes by age, gender, and discipline. I suggest presenting some histograms by gender or discipline, for example. Additionally, the correlations provided by GLMs could be highlighted further throughout the manuscript. I appreciate that the authors have made all data underlying their findings fully available, ensuring transparency and reproducibility.

Even though the authors claim that their findings cannot be extrapolated, I recommend improving the discussion section by adding arguments about the relevance of the studied disciplines. The authors could propose hypotheses about similarities and differences with other disciplines, particularly social sciences. It would be beneficial to include the authors’ thoughts on these results, even if the speculations occasionally extend beyond the current data.

Finally, the conclusions and recommendations are based on the current staff attitudes, but they do not consider attracting individuals with different characteristics. It would be desirable to include thoughts on the complementarity of other higher education policies. How does training interact with recruitment policies? Are there other policies that could help decolonize the curriculum?

6. PLOS authors have the option to publish the peer review history of their article (what does this mean?). If published, this will include your full peer review and any attached files.

Reviewer #1: No

---

## [Author Response · Author response to Decision Letter 0]

19 Sep 2024

Dear Editor Francisca Ortiz Ruiz, Ph.D.

Thank you very much for letting us submit an amended version of our manuscript entitled “Exploring attitudes to decolonising the science curriculum – a UK Higher Education case study”. 

We were very pleased to see the positive and encouraging reviewer’s comments. We have addressed all the comments and included our manuscript amendments in the response to reviewers’ comments. We further provide a manuscript where all amendments appear in red text. 

We look forward to your thoughts on our new, improved manuscript. 

Yours sincerely, 

Lena Grinsted, Catherine Murgatroyd and Jodi Burkett 

Responses to reviewer’s comments 

Reviewer #1: 

Reviewer comment: 

The research analyzes the attitudes towards decolonising the science curriculum among academic teaching staff at a medium-large UK university. There is a lot to like about this work. Although it is not experimental, it presents a valuable case study using a rich dataset from a questionnaire covering demographic characteristics, understanding of decolonization, benefits and risks, training and teaching activities, and the importance and barriers to decolonising the science curriculum. The brief historical background section is also appreciated. Overall, I enjoyed reviewing this manuscript.

Reply: We are thrilled to hear the reviewer enjoyed our manuscript. 

Reviewer comment: 

I find the manuscript to be a technically sound piece of scientific research, with data that support the conclusions and specific recommendations. My primary concern is the lack of explanation regarding the science-related schools under analysis (Biological Sciences, Psychology, Pharmacy, Biomedicine, Geography, and Geosciences). What criteria were used to select these four schools? The central argument of the manuscript is also applicable to social sciences, so an explanation of the selected fields would enhance the discussion and policy implications.

Reply: This is a really good point. As the first author, and as a biologist, I chose to send out the questionnaire to schools that I thought was as similar in approach to my own school of Biological Sciences. We have a breadth of subject areas at our institution, but I was particularly interested in how lecturers in the sciences relate to decolonisation. This is because I, anecdotally, had experienced more resistance from science educators than teachers in other disciplines, with some of my science-colleagues struggling to see the relevance to our discipline. Other schools within our faculty, that I chose not to invite, are more human- and health-focussed. With those subjects I felt that a whole range of additional issues and aspects relating to the legacies of colonisation and racism would be relevant, which was outside the scope of this study. I have now added a sentence in the MS to justify this selection: 

Lines 303-307: “These four school were selected as the focus of this study was on science teaching. The remaining science-related schools at the institution had a more health and medical focus and were not included. This is because healthcare and medicine have a suite of additional challenges relating to the legacies of colonialism and racism due to patient-practitioner interactions [57,66–70].”

Reviewer comment: 

The statistical analysis is performed appropriately and rigorously. The methods are suitable given the small sample size. However, a deeper intersectional analysis could provide further insights into the heterogeneity of attitudes by age, gender, and discipline. I suggest presenting some histograms by gender or discipline, for example. 

Reply: If I understand this comment correctly, then this is what we are already presenting in supplementary materials (Figs S3-S5: Effects of age, gender and discipline on perceptions). 

Reviewer comment: 

Additionally, the correlations provided by GLMs could be highlighted further throughout the manuscript. I appreciate that the authors have made all data underlying their findings fully available, ensuring transparency and reproducibility.

Reply: While we agree that these results would be interesting to discuss in detail, we feel we must be cautious with making conclusions based on these due to our limited sample size within age, gender and discipline categories. However, based on this reviewer comment we have added a sentence in the discussion as follows: 

Lines 710-713: “In our case study, 15-20% of teaching staff did not find it relevant to discuss topics of race and (de)coloniality in their teaching, with a non-significant tendency for older staff and males to rate the importance of, and their own understanding of, decolonisation low.”

Reviewer comment: 

Even though the authors claim that their findings cannot be extrapolated, I recommend improving the discussion section by adding arguments about the relevance of the studied disciplines. The authors could propose hypotheses about similarities and differences with other disciplines, particularly social sciences. It would be beneficial to include the authors’ thoughts on these results, even if the speculations occasionally extend beyond the current data.

Reply: Thank you for this recommendation. Based on your comments we have now amended the introduction and also added a paragraph to the discussion as follows: 

Lines 72-75: “While a case study does not allow us to universalise our findings, we argue that our recommendations are relevant for similar institutions in the UK and beyond who are at a relatively early stage in undertaking a critical review of their curriculum through a decolonial lens.”

Lines 754-775: “While science education in the UK was the focus of this study, we recognise that attitudes to, and progress with, decolonising the curriculum is likely to differ across disciplines, and in other regions of the world. As the authors themselves enjoy academic careers that span the humanities, social and biological sciences, we recognise that knowledge is contextually legitimised and specialised within fields of practice by the actors within them, with varying epistemological and ontological interpretations [82]. This may result in disciplines having different understandings of the concept, practice and application of decolonising, with varying levels of commitment and motivation to de-centre western dominance. We expect that steps towards decolonising the curriculum are more developed in the humanities and social sciences, compared to the life sciences, with a deeper understanding of the concept and less resistance towards it. This is because the decolonisation movement at UK universities started within the humanities, and most research has been published within related fields [8,83–85]. However, while conversations and practices related to decolonising may have been taking place within the humanities and social sciences longer than in the life sciences, they are not without controversy and there is still a lot of work to be done to ensure fully socially just and equitable education and curricula across disciplines in the UK and beyond. The specific requirements and challenges related to decolonising the curriculum might differ between disciplines, but we believe that the trends identified in this case study are illuminating and relevant across disciplines in the life sciences, social sciences and humanities.”

Reviewer comment:

Finally, the conclusions and recommendations are based on the current staff attitudes, but they do not consider attracting individuals with different characteristics. It would be desirable to include thoughts on the complementarity of other higher education policies. How does training interact with recruitment policies? Are there other policies that could help decolonize the curriculum?

Reply: Thank you for this suggestion. We touch upon this topic in the section about barriers to decolonising, that particularly emphasised a lack of time teaching staff experience to address this properly. Our participants also identified that management need to take responsibility and would benefit from training. We mention this is the discussion, but feel it is outside the scope of this paper to expand further upon education and recruitment policies.

---

## [Decision Letter · Decision Letter 1]

10 Oct 2024

Exploring attitudes to decolonising the science curriculum – a UK Higher Education case study

PONE-D-24-23238R1

Dear Dr. Grinsted,

We’re pleased to inform you that your manuscript has been judged scientifically suitable for publication and will be formally accepted for publication once it meets all outstanding technical requirements.

Kind regards,

Francisca Ortiz Ruiz, Ph.D.

Academic Editor

PLOS ONE

Additional Editor Comments (optional):

Dear Author,

Thank you for choosing Plos One for this publication and allowing me to read it.

I suggest accepting this paper, as the reviewer's recommendations have been addressed, and its content has been improved.

Kind regards,

Academic editor

Reviewers' comments:

Reviewer's Responses to Questions

**Comments to the Author**

1. If the authors have adequately addressed your comments raised in a previous round of review and you feel that this manuscript is now acceptable for publication, you may indicate that here to bypass the “Comments to the Author” section, enter your conflict of interest statement in the “Confidential to Editor” section, and submit your "Accept" recommendation.

Reviewer #1: All comments have been addressed

2. Is the manuscript technically sound, and do the data support the conclusions?

Reviewer #1: Yes

3. Has the statistical analysis been performed appropriately and rigorously? 

Reviewer #1: Yes

4. Have the authors made all data underlying the findings in their manuscript fully available?

Reviewer #1: Yes

5. Is the manuscript presented in an intelligible fashion and written in standard English?

Reviewer #1: Yes

6. Review Comments to the Author

Reviewer #1: Thank you for adressing the previous comments. I believe the manuscript is stronger now and it will make a great contribution. I specially liked the last paragraph that you added to the discussion.

7. PLOS authors have the option to publish the peer review history of their article (what does this mean?). If published, this will include your full peer review and any attached files.

Reviewer #1: No

---

## [Editor Report · Acceptance letter]

28 Oct 2024

PONE-D-24-23238R1 

PLOS ONE

Dear Dr. Grinsted, 

I'm pleased to inform you that your manuscript has been deemed suitable for publication in PLOS ONE. Congratulations! Your manuscript is now being handed over to our production team.

Kind regards, 

on behalf of

Dr. Francisca Ortiz Ruiz 

Academic Editor

PLOS ONE